# Integrating Self-Report and Psychophysiological Measures in Waterpipe Tobacco Message Testing: A Novel Application of Multi-Attribute Decision Modeling

**DOI:** 10.3390/ijerph182211814

**Published:** 2021-11-11

**Authors:** Elise M. Stevens, Andrea C. Villanti, Glenn Leshner, Theodore L. Wagener, Brittney Keller-Hamilton, Darren Mays

**Affiliations:** 1Department of Population and Quantitative Health Sciences, Division of Preventative and Behavioral Medicine, University of Massachusetts Medical School, Worcester, MA 01655, USA; 2Vermont Center on Behavior and Health, Department of Psychiatry, University of Vermont, Burlington, VT 05405, USA; andrea.villanti@uvm.edu; 3Gaylord College of Journalism and Mass Communication, University of Oklahoma, Norman, OK 73019, USA; leshnerg@ou.edu; 4Center for Tobacco Research, The Ohio State University James Comprehensive Cancer Center, Columbus, OH 43214, USA; theodore.wagener@osumc.edu (T.L.W.); brittney.keller-hamilton@osumc.edu (B.K.-H.); darren.mays@osumc.edu (D.M.); 5Department of Internal Medicine, The Ohio State University, Columbus, OH 43210, USA

**Keywords:** waterpipe, communication, messaging, psychophysiology

## Abstract

Background: Waterpipe (i.e., hookah) tobacco smoking (WTS) is one of the most prevalent types of smoking among young people, yet there is little public education communicating the risks of WTS to the population. Using self-report and psychophysiological measures, this study proposes an innovative message testing and data integration approach to choose optimal content for health communication messaging focusing on WTS. Methods: In a two-part study, we tested 12 WTS risk messages. Using crowdsourcing, participants (*N* = 713) rated WTS messages based on self-reported receptivity, engagement, attitudes, and negative emotions. In an in-lab study, participants (*N* = 120) viewed the 12 WTS risk messages while being monitored for heart rate and eye-tracking, and then completed a recognition task. Using a multi-attribute decision-making (MADM) model, we integrated data from these two methods with scenarios assigning different weights to the self-report and laboratory data to identify optimal messages. Results: We identified different optimal messages when differently weighting the importance of specific attributes or data collection method (self-report, laboratory). Across all scenarios, five messages consistently ranked in the top half: four addressed harms content, both alone and with themes regarding social use and flavors and one addiction alone message. Discussion: Results showed that the self-report and psychophysiological data did not always have the same ranking and differed based on weighting of the two methods. These findings highlight the need to formatively test messages using multiple methods and use an integrated approach when selecting content.

## 1. Introduction

Waterpipe (i.e., hookah) tobacco smoking (WTS) is one of the most prevalent types of smoking among young people in the United States [1], and nearly 30% of young non-users are susceptible to use [2]. Because WTS produces harmful toxicants [3,4], which can cause cancer and other negative health outcomes [5,6], it is imperative to communicate the risks of use to the young adult population. Despite the need for risk communication and the proliferation of social media messaging promoting WTS smoking [7,8], there is little public education on WTS [2,9,10,11,12,13,14,15,16]. The 2016 Deeming Rule gave the U.S. Food and Drug Administration (FDA) regulatory authority over all tobacco products, including waterpipe tobacco [17]. Now, the FDA has the opportunity to disseminate public health education campaigns to effectively communicate the risks of WTS as they have with cigarettes [18] and e-cigarettes [19]. However, there is little research on how to effectively communicate the risks of WTS to young people [9].

The existing research base examining WTS messaging has relied heavily on self-report data assessed after message exposure [13,20], with few exceptions to that method [21]. Testing messages using only self-report data in formative work is the norm in tobacco prevention mass media campaign development [22]. For example, the FDA pre-tests campaign messages using self-report measures that include perceived message effectiveness, smoking-related beliefs, and attitudes after exposure to determine optimal content [23]. Because self-report measures can be limited by social desirability, response bias, and inaccurate recall, they can be complemented by psychophysiological assessments during message exposure to gain more insight into how individuals will react to messages in real time [24]. Psychophysiological measures such as heart rate combined with recognition tasks and eye-tracking provide measures of message processing and responses that are not directly accessible through self-report [25,26]. This includes cognitive resources allocated to message processing through heart rate measurement, encoding of the message through recognition tasks, and visual attention—processes that have been linked to changes in behavioral attitudes and intentions [27].

Psychophysiological measurements rely on the foundational premise of the Limited Capacity Model of Motivated Mediated Message Processing (LC4MP) [28], which addresses humans’ innate ability to respond to messaging emotionally and cognitively but with only a limited number of resources to encode, store, and retrieve the information later. Humans have two motivational systems that are activated when confronted with a message: appetitive/approach and aversive/avoid [28]. Psychophysiology allows researchers to collect these motivational reactions to complement self-report measures in order to understand which messages are encoded and stored at an increased rate in relation to other messages [28]. With only self-report data, researchers may not understand why some messages are leading to high aversion and low encoding and storage [29]. Heart rate, recognition tasks, and eye-tracking provide the ability to capture processes in real time during message exposure to ensure content is optimized for cognitive encoding and storage in the formative stage. For instance, heart rate deceleration indicates more cognitive resources allocated to message encoding [24]. Increased recognition indicates encoding of information [24], and eye-tracking ensures individuals are dedicating substantial visual attention to the message and captures attention to specific content (i.e., areas of interest) as well. The addition of psychophysiological data to campaign formative research can provide insight into message processing [24] and help optimize campaign content targeting health-related beliefs and behaviors. These psychophysiological methods coincide with cognitive and emotional responses that correlate with behavior change [24]. However, a significant limitation of including psychophysiological methods in campaign development has been the lack of methods to integrate psychophysiology data with the standard self-report data collected during the formative phase.

Multi-attribute decision-making (MADM) models provide a practical, transparent framework for considering data across multiple inputs (attributes), determining the relative importance of each attribute, and simultaneously evaluating detail at the attribute-level and in a summary measure across all attributes using aggregate data [30]. They have been widely used in business [31], economics [32], technology [33], and health care [34], but rarely in health communication science. Multi-attribute models are a decision analytic technique that allows for inclusion of qualitative and quantitative factors in decision-making [30] as is the case with multiple attributes for each message across multiple messages tested. The model evaluates each message by assigning each attribute a raw score obtained from data collection for each message, determining the relative importance (weight) of each attribute in the decision-making, creating a summary score for each message across attributes, and assigning an overall score or ranking to each message. Identifying the key attributes and their relative importance at the outset of decision-making allows for optimizing desired outcomes, using an algorithmic approach to combining data across sources, and conducting analyses to determine the impact of various attributes on overall rankings of the messages considered. This type of framework allows for explicit consideration and prioritization of specific message outcomes or specific types of data in determining optimal message content for a health communication campaign.

This study used a MADM approach to choose optimal WTS public health messages for young adults [30] by integrating traditional self-report measures and lab-based heart rate, eye-tracking, and recognition tasks. Rooted in LC4MP, a recent meta-analysis of self-report, behavioral, and psychophysiological responses to messages indicates self-report outcomes capture the largest effects, followed by behavioral and psychophysiological measures. Traditionally, these measures have been considered separately to inform message development. To our knowledge, they have not been systematically combined to optimize message content, and based on variability in effect size estimates [35], there has been no research to examine whether these methods and individual measures should be considered equally, or if certain methods or measures should be weighted more substantially, in making decisions about optimal messages. This has important practical implications for the development of public health campaigns, as practitioners who develop campaigns face decisions about the optimal message content to use relative to campaign resources. Integrating self-report measures of receptivity, engagement, attitudes, and emotions combined with psychophysiological measures of heart rate, eye-tracking, and a recognition task through an in-lab experiment, this study aimed to demonstrate the utility of this data collection approach and provide a roadmap for determining the optimal messaging content across multiple data sources and attributes using MADM.

## 2. Materials and Methods

### 2.1. Participants

Participants in both the crowdsourced testing and the in-lab study met the following criteria: (1) 18–30 years old, (2) self-reported WTS in the past month or had never smoked waterpipe tobacco but were deemed susceptible [36], and (3) within the United States. Participants were deemed as susceptible if they gave a response other than “No” to one or more questions about WTS soon, in the next year, sometime in the future, or if a best friend offered it [36].

### 2.2. Message Development and Selection

Using information from multiple contexts including research on health harms and addictiveness [37], health communication studies examining risk messaging [13,38], past WTS message testing studies [10,11,39], and young adults’ beliefs about WTS [40] and flavors [41,42], we created a bank of 18 candidate messages. Next, we consulted experts in the field using an iterative process until consensus was reached on the messages to use. Four key themes for effective hookah messaging emerged: (1) health harms, (2) addictiveness, (3) social use, and (4) flavoring. To reflect this, the current study resulted in a 2 (risk content: health harms or addiction) by 3 (message theme: risk alone; flavors; or social use) design. The 6 experimental conditions were: harms risk alone; harms, flavors; harms, social use; addiction risk alone; addiction, flavors; and addiction, social use. All messages were consistent in length (i.e., two sentences with one hashtag), layout, and structure, and only differed on their manipulation of risk content and message theme. Text content was accompanied by 1 of 4 counterbalanced images relevant to WTS use: a waterpipe; social WTS setting; a smoke cloud; and a ring of smoke. All messages were pre-tested for readability. See Table 1.

### 2.3. Crowdsourced Testing

#### 2.3.1. Setting

In September 2018, participants (*N* = 713) were recruited using the crowdsourcing platform, Amazon Mechanical Turk (MTurk). We used data quality assurance measures including verification requirements to prevent automated responses from bots and prohibiting duplicate responses through manual review of the data. In addition, we used randomly generated completion codes [43,44].

#### 2.3.2. Design

After meeting eligibility criteria and providing online consent, participants answered questions assessing demographics, WTS and other tobacco use, and intentions to engage in WTS. In a 2 (user status: current hookah user, susceptible non-user) by 2 (risk content: health harms or addiction) by 3 (message theme: harms/addiction risk alone, harms/addiction risk flavors, or harms/addiction risk social use) design, participants were randomly assigned to one of the six risk content by message theme categories and presented with a single message within that category. After viewing the candidate message for as long as they wanted, participants were asked questions pertaining to the study outcomes. Participants were compensated $2 and took on average 10 min to complete the survey. All procedures were approved by the host institution’s institutional review board. Appendix A presents details on the messages tested. Procedures have been published elsewhere [20].

#### 2.3.3. Measures

Receptivity: Message receptivity was assessed with 9 items, such as “The message grasped my attention” and “The message was convincing”. Each item was rated on a 7-point scale from 1 (strongly disagree) to 7 (strongly agree; Cronbach’s α = 0.92) [38,45].

Engagement: Engagement was assessed with 3 items asking about how engaged, how involved the participant was in the message, and how much attention was paid to the message on a scale from 1 (none) to 7 (very; Cronbach’s α = 0.94).

Positive attitude: Positive attitude toward the message was assessed with 9 items on semantic differential scales from 1 to 7. Examples included “boring/exciting”; “not stimulating/stimulating”, with higher scores reflecting a more positive attitude toward the message (Cronbach’s α = 0.94) [36].

Negative emotions: Negative emotions were assessed with 4 items: frightened, anxious, nervous, and worried in response to the message on a scale from 1 (not at all) to 4 (extremely; (Cronbach’s α = 0.91) [46].

### 2.4. Psychophysiological Testing

#### 2.4.1. Setting

Participants (*N* = 120) were recruited through mass emails and social media advertising at a large southwestern university. Eligible participants completed a one-hour in-person laboratory visit. Participant characteristics were similar in both studies. Demographics have been reported elsewhere [20,21].

#### 2.4.2. Design

Once eligibility was determined, participants were scheduled for a lab visit. Upon arriving to the lab, written consent was obtained, baseline measures identical to those in the crowdsourcing survey were assessed, and then participants were randomly assigned to view 12 messages on combinations of each of the two risks with one of three themes: (a) health harms or addictiveness alone, (b) health harms or addictiveness with flavors, and (c) health harms or addictiveness with social use (Appendix A). Messages were presented in a random order. During the viewing of the messages, participants were monitored for heart rate and eye-tracking. After completing the viewing task, participants completed a recognition task where they indicated whether an excerpt of the message came from the messages they just saw or if they were foils. The full description of the psychophysiological testing is available elsewhere [21]. All procedures were approved by the host institution’s institutional review board.

#### 2.4.3. Measures

Heart rate: Heart rate was measured to assess the cognitive resources allocated to the message [24]. Using four gelled sensors applied to the participants’ forearms (2) and left wrist (2), electrocardiogram (ECG) signals were recorded with a Shimmer 3 EXG module, while viewing the messages. Raw data were sampled at 512 Hz. Heart change scores were computed using beats per minute (BPM) and subtracting the last second of the baseline (BPM prior to message viewing) from scores for each second of message exposure for each participant. Heart rate scores were set to absolute values, but all reflect the amount of deceleration. Then, scores were averaged across the message, giving each message its own score. Greater values indicated greater deceleration in heart rate, signaling more cognitive resources allocated to the message [24].

Recognition accuracy: To assess recognition and the degree to which the message content was encoded into short-term memory, participants completed a recognition task. The task consisted of 48 fragments shown to participants in which they had to click “yes” or “no” as to whether it appeared in the messages they had just viewed. The fragments included 24 targets taken from the messages and 24 foils. Recognition accuracy was computed as a percentage of hits (i.e., responding “yes” to targets) [47].

Visual Attention: Visual attention was assessed using an eye-tracker connected to the base of the computer screen. For each message, three areas of interest (AOI) were identified—one for each of the two sentences and one for the hashtag included. All three AOIs were averaged for the number of milliseconds that a participant viewed the AOIs. Visual attention is reported in absolute values.

### 2.5. Data Integration and Analysis

Self-report scores for each construct (i.e., receptivity, engagement, positive attitude, and negative emotion) were averaged for each participant, and then averaged across participants to create a mean score for each message. Psychophysiological scores for each construct (i.e., heart rate deceleration, recognition accuracy, visual attention) were also averaged across participants to create a mean score for each message. Within each attribute, mean scores were ranked across messages, with the highest scores receiving the lowest ranking (i.e., “1”; most effective) and the lowest scores receiving the highest ranking (i.e., “12”; least effective).

Using the MADM framework [30,48], we created a decision matrix in which each row (e.g., alternative or message) represented a message and each column (attribute) represented a construct collected via the crowdsourced or psychophysiological study. For each study, the rank of each attribute was summed within each message to create a total rank for each message by study type (self-report, psychophysiological).

We used the attribute-level ranks for each message to compute summary scores and rankings under four weighting scenarios to identify the top-performing messages: (1) Equal attribute weights (4 self-report, 3 psychophysiological); (2) Equal study weights (Total self-report rank, total psychophysiological rank); (3) Prefer self-report (Individual self-report attribute ranks, total psychophysiological rank); and (4) Prefer psychophysiological (Total self-report rank, individual psychophysiological attribute ranks).

## 3. Results

### 3.1. Self-Report Data and Rankings

Table 1 shows mean scores for the crowdsourced self-report measures of receptivity, engagement, positive attitude, and negative emotion for each message. While ranking by self-report attribute differed across WTS messages, the summary rankings identified the top performing messages in the harms alone category (H1: “Many people think smoking hookah is safer than cigarettes. Truth is, hookah tobacco has the same health effects—breathing problems, cancer, lung and heart disease. #UnfollowHookah”), followed by harms/social category (HS2: “Consider skipping your turn. Sharing a hookah hose increases your risk of infections like herpes. #UnfollowHookah”), followed by harms/flavors category (HF2: “Don’t let flavored hookah tobacco smooth talk you. Flavored hookah tobacco masks the nasty chemicals and is manufactured the same way as cigarettes. #UnfollowHookah”), and followed by harms/social category (HS1: “Socializing at hookah bars isn’t all fun and games. Smoking hookah with friends exposes you to even more poisonous chemicals than cigarette smoking. #UnfollowHookah”).

### 3.2. Psychophysiological Data and Rankings

Mean heart rate deceleration, recognition accuracy, and visual attention scores are presented by message, with their attribute-level rankings in Table 1. Again, ranking by psychophysiological attribute differed from the summary rank for all psychophysiological measures combined. Summary rankings based on the psychophysiological data identified the top performing measures in the harms alone (H2: “The nicotine buzz comes with more than you bargained for. ‘Hookah sickness’ can be carbon monoxide poisoning. #UnfollowHookah”), addiction/flavor (AF1: “Mint, Blue Mist, Apple, Ambrosia. Flavors mask the bitter taste of nicotine and help get you hooked on smoking hookah. #UnfollowHookah”), and harms/flavor categories (HF2: “Don’t let flavored hookah tobacco smooth talk you. Flavored hookah tobacco masks the nasty chemicals and is manufactured the same way as cigarettes. #UnfollowHookah”).

### 3.3. Weighting Scenarios Integrating Self-Report and Psychophysiological Data

Table 2 presents the mean score and rank of each message under four weighting scenarios. In the equal attribute ranks scenario, each of the four self-report measures and 3 lab measures received equal weight in the mean score. Results from this method identified the top three messages in the harms/flavor (HF2), harms/social (HS2), and harms alone (H1) categories. The equal study weights averaged the two total ranks (crowdsourced, psychophysiological) for each message, selecting harms/flavor (HF2), harms alone (H2), and harms/social (HS2), and as the top performing messages. The third scenario provided greater weight to the self-report measures, using the four attribute-level ranks for the self-report measures and the total rank for the psychophysiological measures; the top three messages identified using this message were the harms/flavor (HF2), harms/social (HS2), and harms alone (H2). The final scenario provided greater weight to the psychophysiological measures, using the total rank for the self-report measures and the three attribute-level ranks for the psychophysiological measures. Similarly, this scenario identified harms/flavor (HF2) as the top performing message, followed by harms/social (HS2), and addiction/flavor (AF1).

## 4. Discussion

The current study tested a novel methodology to harness both self-report and lab-based data from formative message testing to optimize the selection of messages for a WTS education campaign. Summary findings identified different optimal messages when considering different scenarios for weighting the importance of specific attributes or study design. Across all scenarios, five messages consistently ranked in the top half: HF2 (“Don’t let flavored hookah tobacco smooth talk you. Flavored hookah tobacco masks the nasty chemicals and is manufactured the same way as cigarettes. #UnfollowHookah”), HS2 (“Consider skipping your turn. Sharing a hookah hose increases your risk of infections like herpes. #UnfollowHookah”), H2 (“The nicotine buzz come with more than you bargained for. ‘Hookah sickness’ can be carbon monoxide poisoning. #UnfollowHookah”), H1 (“Many people think smoking hookah is safer than cigarettes. Truth is, hookah tobacco has the same health effects—breathing problems, cancer, lung and heart disease”. #UnfollowHookah”), and A1 (“No one is invincible. Even smoking hookah just once or twice a month can lead to nicotine addiction. #UnfollowHookah”). Of these messages, four addressed harms content, both alone and with message themes regarding social and flavored use. Generally, the addiction content messages performed worse than the harms content messages, with the exception of the addiction alone message (A1) and one addiction/flavor message (AF1: “Mint, Blue Mist, Apple, Ambrosia. Flavors mask the bitter taste of nicotine and help get you hooked on smoking hookah. #UnfollowHookah”) that received a high ranking in “prefer lab” weighting scenario. The latter message (AF1) was ranked low (9th) in the self-report data, but performed well with respect to heart rate deceleration, recognition accuracy, and visual attention. These findings align with other work on self-reported measures for vaping prevention message testing that identified harms content as more effective than addiction content, though they differ with respect to the social and flavor themes [49]. Whereas rankings in the current study identified harms/social and harms/flavor messages for WTS as some of the most effective, our other research on vaping prevention messages identified them as potentially eroding the effectiveness of the harms messages [49]. With regard to the addiction findings, contrary to other addictive substances (i.e., heroin), addiction to WTS might not carry the same social stigma and therefore an addiction message might not be as impactful for this behavior. Future research could test messaging strategies such as the long-term financial burden of continued use to address WTS addiction. Finding ways to effectively communicate addiction risk is imperative for this young population.

Previous work assessing the effectiveness of marijuana prevention messages using self-report and psychophysiological data identified the latter as providing a more consistent picture of message processing [50]. However, few if any communication campaigns use these procedures in message development. In our current analyses, focusing on one method over the other (i.e., self-report versus psychophysiological) would have resulted in the selection of different messages for a WTS campaign. Our data suggest that a campaign derived from self-report data alone would have focused on harms content, while a campaign focused on psychophysiological data would have a greater focus on addiction content. This may partially explain difficulties in creating effective addiction content in substance use prevention campaigns, as reliance on self-reported outcomes in the formative stage may deter researchers from using addiction content.

Findings from our analyses using different weighing scenarios support the need to formatively test messages in multiple ways and use an integrated approach when selecting content for media campaigns. A recent meta-analysis of the LC4MP indicated that psychophysiological measurement can be low signal, high noise [24], but self-report and behavioral outcomes often provide larger effect sizes [35]. Because of this, an integrated approach using both self-report and psychophysiology is beneficial. The methods used in this study were easy to implement and allowed our team to explore a range of potential campaigns based on the range of data we collected during our message testing phase. Ultimately, we used the results from the equal attribute ranks scenario to select four messages (HF2, HS2, H1, A1) to include in a randomized controlled trial on WTS risk education (NCT04252014). Results from the trial are forthcoming. Based on our analyses, researchers and practitioners applying this multi-method approach could use a similar strategy to giving equal weights to self-report and psychophysiological attributes to inform message optimization decisions.

The limitations of this study include not testing this approach on actual effectiveness and behavior change. Research testing the various weighted scenarios with a behavioral outcome could further shed light on which weighted approach is predictive of behavior change. In addition, we used Mturk for crowdsourced data collection, which may not be representative of the target population for our messaging. However, past research has shown that Mturk samples used for tobacco research produce comparable experimental and correlational results to those of population-based samples [51]. The study also was not powered to detect how use of other nicotine and tobacco products impacted outcomes. Future studies should consider other tobacco and nicotine use status. Lastly, this study only tested WTS messages, and others should test this method with other health behavior messages. The MADM analytic approach we used does not test for statistical differences between messages, and in the future it will be prudent to pair this method with analyses that test hypotheses about message effects statistically [20,21]. Despite these limitations, this study was the first of its kind to use a MADM to determine optimal message content in the formative pre-testing of health communication messaging and makes an important contribution to the future of health communication research.

## 5. Conclusions

Integrating self-report and psychophysiological data using MADM is an innovative strategy to choose optimal content when designing health behavior messaging. By utilizing data from various inputs and following the decision-making model, messages chosen for a mass media campaign may induce a larger impact on public health targets. This strategy may be the most useful in formative work, when pre-testing messages to garner more information about how the campaign messages may fare among the target audience. Our analysis ultimately pointed to using equal attribute weights for the self-report and psychophysiological measures to identify optimal message content; however, future studies should continue to test and refine this approach. Future studies should continue to test and refine this approach in tobacco prevention messaging and in other health content areas. There is also a need for studies to compare the impact of self-report vs. psychophysiological data or integration of the two in selection of campaign messages on campaign-targeted beliefs and behaviors. These types of studies may shed light on how best to apply MADM in the development and evaluation of health communication campaigns.

## Figures and Tables

**Table 1 ijerph-18-11814-t001:** Multi-Attribute Decision Matrix for Crowdsourced and Psychophysiology Waterpipe Tobacco Smoking Message Testing.

	Crowdsourced (Self-Report)										Psychophysiological (Lab)							
	Receptivity		Engagement		Attitude		Negative Emotion		Total		Heart Rate Deceleration		Recognition Accuracy		Visual Attention		Total	
Message	Mean (SE)	Rank	Mean (SE)	Rank	Mean (SE)	Rank	Mean (SE)	Rank	Sum of Attribute Ranks	Rank	Mean (SD)	Rank	Mean (SD)	Rank	Mean (SD)	Rank	Sum of Attribute Ranks	Rank
H1	5.29 (0.16)	1	5.77 (0.14)	4	4.86 (0.18)	3	2.20 (0.11)	2	10	1	2.82 (0.50)	3	0.871 (0.03)	4	1220.72 (730.11)	12	18	12
H2	4.92 (0.19)	6	5.94 (0.16)	1	4.53 (0.20)	7	2.04 (0.12)	3	17	5	2.87 (0.57)	5	0.677 (0.03)	3	1375.4 (779.43)	1	14	1
HS1	5.15 (0.19)	2	5.74 (0.16)	6	4.85 (0.20)	4	2.04 (0.12)	3	15	4	2.45 (0.57)	6	0.698 (0.03)	6	1287.5 (786.94)	10	25	10
HS2	5.04 (0.17)	4	5.78 (0.15)	3	4.98 (0.19)	2	2.23 (0.11)	1	10	1	2.16 (0.47)	8	0.878 (0.02)	8	1298.62 (763.33)	6	15	6
HF1	4.89 (0.18)	7	5.74 (0.15)	6	4.64 (0.19)	5	1.99 (0.12)	6	24	7	2.24 (0.51)	7	0.765 (0.03)	7	1293.23 (765.70)	8	22	8
HF2	5.10 (0.19)	3	5.77 (0.16)	4	5.01 (0.20)	1	2.01 (0.12)	5	13	3	2.60 (0.52)	4	0.798 (0.03)	5	1335.34 (796.33)	3	12	3
A1	5.01 (0.17)	5	5.81 (0.14)	2	4.54 (0.18)	6	1.88 (0.11)	8	21	6	1.65 (0.51)	10	0.769 (0.03)	10	1317.9 (781.80)	4	20	4
A2	4.71 (0.16)	8	5.63 (0.14)	10	4.51 (0.18)	9	1.99 (0.11)	6	33	8	1.98 (0.47)	9	0.803 (0.03)	9	1289.96 (781.45)	9	21	9
AS1	4.50 (0.18)	11	5.56 (0.16)	11	4.53 (0.20)	7	1.81 (0.12)	10	39	9	1.64 (0.45)	11	0.517 (0.04)	11	1317.12 (764.58)	5	28	5
AS2	4.10 (0.17)	12	5.73 (0.14)	8	4.23 (0.18)	12	1.73 (0.11)	12	44	12	2.88 (0.51)	2	0.643 (0.03)	2	1295.01 (745.07)	7	20	7
AF1	4.57 (0.17)	10	5.38 (0.15)	12	4.47 (0.19)	11	1.83 (0.12)	9	42	11	3.03 (0.52)	1	0.798 (0.03)	1	1344.93 (806.43)	2	7	2
AF2	4.63 (0.18)	9	5.64 (0.15)	9	4.48 (0.20)	10	1.74 (0.12)	11	39	9	1.47 (0.43)	12	0.727 (0.03)	12	1261.06 (752.70)	11	31	11

Note: Message receptivity was assessed on a 7-point scale from 1 (strongly disagree) to 7 (strongly agree). Engagement was assessed on a scale from 1 (none) to 7 (very). Positive attitude toward the message was assessed on semantic differential scales from 1 to 7. Negative emotions were assessed on a scale from 1 (not at all) to 4 (extremely). Visual attention is measured in milliseconds.

**Table 2 ijerph-18-11814-t002:** Ranking of waterpipe tobacco smoking messages by weighting scenario.

	Equal Attribute Ranks (4 Self-Report, 3 Lab)		Equal Study Weights (Total Self-Report Rank, Total Lab Rank)		Prefer Self-Report (Individual Self-Report Attribute Ranks, Total Lab Rank)		Prefer Lab (Total Self-Report Rank, Individual Lab Attribute Ranks)	
Message	Summary Score	Rank	Summary Score	Rank	Summary Score	Rank	Summary Score	Rank
H1	4.0	3	6.5	5 *	4.4	4	4.75	4 *
H2	4.4	4	3.0	1 *	3.6	3	4.75	4 *
HS1	5.7	5	7.0	7 *	5	5 *	7.25	7 *
HS2	3.6	1 *	3.5	3	3.2	1 *	4.00	2
HF1	6.6	7	7.5	9	6.4	7	7.25	7 *
HF2	3.6	1 *	3.0	1 *	3.2	1 *	3.75	1
A1	5.9	6	5.0	4	5	5 *	6.50	6
A2	7.7	9	8.5	10	8.4	8	7.25	7 *
AS1	9.6	11	7.0	7 *	8.8	9 *	9.25	11
AS2	9.1	10	9.5	11	10.2	12	8.00	10
AF1	7.0	8	6.5	5 *	8.8	9 *	4.50	3
AF2	10.0	12	10.0	12	10	11	10.00	12

Note: * denotes tied ranking.

## Data Availability

Data available on request.

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
