# Peer review of "Integrating Self-Report and Psychophysiological Measures in Waterpipe Tobacco Message Testing: A Novel Application of Multi-Attribute Decision Modeling"

_ijerph, 2021, doi:10.3390/ijerph182211814_

Round 1

Reviewer 1 Report

An interesting paper testing a new methodology with potential to improve the effectiveness of health promotion campaigns across a range of topics and behaviours extending beyond the tobacco/nicotine theme used in this experiment.

I don’t have any major critique of the concept or the methods, but I do have a couple of points which I’d like the authors to consider in relation to the discussion and conclusion.

The authors mentioned that the addiction content messages performed worse that the harms content messages. This may not be because people feel more strongly about harms than addiction, rather it may be related to the way the messages have been framed for use in this experiment.

The concept of addiction tends to have more social stigma when used in relation to illegal drugs/hard drugs/drugs perceived as dangerous and life threatening in the short term. When using the concept of addiction in relation to a drug that is legal, cheap, and sold in supermarkets and corner stores, and available in burnt, snorted, chewed or vaporized forms freely and easily, and even promoted as ‘sexy’ or ‘cool’, that kind of addiction may not be considered as socially stigmatizing. For example, many people are addicted to caffeine via their consumption of coffee, but they don’t feel embarrassed by it, and nicotine addiction via a “social and fun activity” like WTS might be considered similar. However, those same people might feel significant shame if they had to publicly admit to heroin addiction, as is it illegal and carries high social stigma.

Given this perspective, the results around addiction messaging might have looked different if the messaging had instead focused on the concept of lifelong involuntary dependence on a drug and the lifelong financial burden of tobacco addiction (with some dollar figures for reference – for example, the cost of a new car being spent on nicotine over X-years), alongside the obvious health harms this brings.

Also, given the sample was drawn from university students (mostly), their relative youth can influence their concept of long-term addiction (in that it can be hard for a person in their early twenties to imagine themselves being 40 or 50 and unable to quit nicotine), which may further reduce the impact of addiction messaging.

Reviewer 2 Report

This paper attempts to address an important issue – how to select the best (ultimately, most effective) messages based on formative evaluation. Formative evaluations primarily rely on self-report, which is a serious shortcoming that has been extensively described. Other measures, such as psychophysiological measures used in this paper, can augment and aid in determining the candidates for best messages. The question of how to combine the different types of measures (and if it should be done) is a valid one. However, the way this paper attempts to answer this question leaves much to be desired.

Here are a few major issues with this paper.

Theoretical contribution: The ultimate goal of the formative evaluation is to select most promising messages for the overall campaign that can then affect behavior. As such, the goal of using the MADM approach and running the analyses in this paper is to provide a means to use different self-report and psychophysiological measures to select these messages – those that will have a desired effect on behavior. However, this paper provides very little in terms of linking its message selection approach to the ultimate behavior. Perhaps it would be useful to mention how psychophysiological measurements are related to what we are ultimately interested in – behavior.

Related to the previous point, the ranking systems do not seem to be theory-derived and this section reads just as an exercise of 'let’s throw things together and see what happens.' Given the slightly different scores it is not surprising that giving different weights results in different outcomes. Based on the reading about MADM in the introduction I was expecting a more theoretical (or at least thought-out) approach to the assumptions around weighting.

Data analysis – averaging scores across participants removes a lot of nuances. While it might be fine for the self-report measures where each participants evaluated only one message, it is not appropriate for the physiological measures where multiple messages were assessed by a single participant. A better approach would be to nest the individual messages within participants or do mixed models.

While ranking is fine, what is more useful to know is whether the differences between 1st and 2nd highest rated messages are meaningful or if they are so small that either message would be appropriate to use. Is the difference of 31 milliseconds really meaningful?  Similarly, the difference between the lowest emotional score (1.73) and the highest (2.23) are not big either.

The messages are very similar in execution and text, so it is not surprising that they are evaluated very similar to each other. It is questionable whether the exercise in correlating the ranking scores (which might be due to chance rather than meaningful difference) provides any useful contribution.

The authors do a good job citing their own past work but they leave out the extensive work on message testing in tobacco area by other teams.

It is not surprising that AF1 was highly recognized because it is very specific. Does it mean that it was encoded better and if yes, how would this ultimately translate into behavior?

Minor issues:

Introduction: “Using only self-report measures may be the underlying reason that campaigns can fall short or produce undesirable effects, such as avoidance or resistance to the message.[21]” There are a few issues with this statement. First, the claim that campaigns fall short or produce undesirable effects is not supported. The citation that the authors use is appropriate, but even the cited article itself discusses that the occasions when fear “campaigns fall short or produce undesirable effects” are very rare. If the authors want to make this point, they need to pick an article that EMPIRICALLY shows that the campaigns do produce undesired effects (but, at least based on the meta-analysis by Tannenbaum et al 2015, these instances of backfiring are very hard to find). Then they can make a supposition that this failure might be due to self-report measures (and if they do that, it needs to be clearly explained how self-report measures are linked with possible campaigns failure). It might be easier to just criticize self-report measures based on other issues, without resorting to the tenuous link with failed campaigns (which itself is a widely discussed, but not a well-documented phenomenon).

In explaining Multi-Attribute Decision-Making model, it would be helpful to state from the start whether this model focuses on individual decision-making (i.e., do viewers themselves employ some sort of algorithm) or if it is primarily used to analyze data collected from the individuals.

Methods:  please define your criteria for “susceptible” individuals.

Please provide more details regarding “data quality assurance measures” used in the MTurk sample.

Was the study approved by an IRB? This information needs to be provided as well as the details regarding consent (was it written consent, waiver of documentation of consent, etc.)?

Were these messages pretested for readability? I had a hard time reading the text on the multi-colored background.

More information on the crowd-sourcing study needs to be provided (even if it was reported in a different publication) – were participants compensated? How long did the study take? How long did participants view each message?

Recognition accuracy – it is unclear how the measures of “recognition accuracy” and “recognition sensitivity” were combined. (And it is confusing to have one of sub-measures labeled the same as the overall measure.)

Were visual attention measures reported in absolute terms? Or relative to the time spent looking at the whole message (normalized)? It is unclear.

Table 1 needs to specify the measurement scale for each item so the readers don’t have to go back to the text to remind themselves whether something was on 1-7 scale or 1-5 and what the numbers for the visual attention stand for.

SD should be reported for visual attention measures.

Reviewer 3 Report

Abstract: It is not clear that messages were framed in a 3x2 so it is difficult to interpret the results listed

Abstract: In the sentence "Across all scenarios, five messages consistently ranked in the top..." include that one addressed addiction alone

Materials & Methods: When listing the 2 (health harms or addiction) x 3 (risk alone, flavors, social use), can just list "risk alone, flavors, and social use" - including "harms/addiction" is cumbersome and confusing with respect to the design.

Materials & Methods: Would appreciate more information on what makes a person susceptible and why 18-30 was the selected age range (esp. why not younger persons who may be highly susceptible to alternative nicotine delivery systems.

Materials & Methods (Message Development & Selection): Reference Table 1 when describing the messages

2.3.3. Measures: "Engagement" section, it is unclear what the question regarding "how much involved" might look like, could you please clarify?

2.4.1. Setting: Might be helpful to show a demographic breakdown of the sample for MTurk and for the second portion of the study. Were they the same demographically? If not, may be a consideration for the discussion section.

3.2. Psychophysiological data and rankings: Unclear why H2 and AF1 are described but HF2 is listed as "seen above"

Table 2: Consider denoting tied scores to promote easy digestion

Discussion: In lines 285-297, some conflicting results are presented, do the authors have any elaboration as to why those findings might have differed?

Discussion: Just a note, if psychophysiological data presents high noise and low signal, might it be inappropriate to weight based on it? Would it perhaps have been prudent to use the same sample for effectiveness and psychophysiological testing?

Discussion: Line 319, missing ending comma

Round 2

Reviewer 2 Report

The authors did a nice job responding to the minor issues. However, some of the major issues raised in the first round have not been addressed sufficiently.  

The issue of selecting different weights based on theory has not been resolved. Adding the sentence about LC4MP does little in explaining how it informed the selection of the weighting approaches. The paper still embodies 'let’s throw things together and see what happens' approach and provides little guidance to how select the weights. I might be confused because in a couple of places using different weighting approaches is referred to as “sensitivity analysis”. (e.g., “Findings from our sensitivity analyses using different weighing scenarios support 330 the need to formatively test messages in multiple ways and use an integrated approach 331 when selecting content for media campaigns.”) I understand sensitivity analyses as complementary to primary analysis, but here it is unclear which analysis (i.e., weighting approach) was primary. Something needs to be said about which weighting mechanism needs to be ultimately selected. The conclusion calls for “Integrating self-report and psychophysiological data using MADM”, but should the readers who follow this call do the same thing and try four (or more) different weighted approaches and select messages that come on top most frequently? Or should they use one of the weighted approaches? The explicit recommendation needs to be included in the paper.

Regarding whether the difference between ranks is meaningful or not, it would be useful to include in the discussion or the limitations something along the lines that the authors provided in their response that this approach does not provide information on the actual differences in the message effects and discuss the implications of that.

p.8, line 204 – should “and sensitivity” be removed? (Seems like it was left behind despite the response “This was an error. We have included the score for only recognition accuracy. We have removed the parts of the sentence pertaining to sensitivity.”)
